# From Large to Powerful: International Comparison, Challenges and Strategic Choices for China's Livestock Industry

Zizhong Shi [1,†] , Junru Li [2,†] and Xiangdong Hu [1,*]

1   Institute of Agricultural Economics and Development, Chinese Academy of Agricultural Sciences, Beijing 100081, China; shizizhong@caas.cn
2   College of Economics and Management, China Agricultural University, Beijing 100083, China; lijunru901@163.com
*   Correspondence: huxiangdong@caas.cn
†   These authors contributed equally to this work.

**Abstract:** Accelerating the construction of a strong livestock industry is of great significance to better guarantee the supply of livestock products and improve the quality, efficiency and competitiveness of the industry. This study constructed an evaluation index system including supply security, scientific and technological support, industrial resilience and international trade to evaluate the strength of China's livestock industry, and then conducted an in-depth analysis of the issues and challenges of the construction of a livestock powerhouse. The research results showed that China's livestock industry ranked 5th in the world, such that China was transitioning from being a large-producing livestock country to being a livestock powerhouse, although improvement was still needed to reach the goal. There were significant differences across species. China's layer industry was a world leader; the pig, sheep and goat, and broiler industries were strong; and the beef cattle and dairy industries were weaker. There are still many challenges, such as the fact that the domestic supply security capacity needs to be strengthened, the level of scientific and technological support needs to be improved, the modern operation system needs to be sound, the industry and supply chain are not highly resilient, the international trade risks are increasing, and the policy support system needs to be improved. It is recommended to improve the institutional mechanism for the construction of the livestock industry, promote a high level of self-reliance and self-improvement in science and technology, build a modern livestock operation system, enhance the resilience and security level of the industry and supply chain, and consolidate and expand international trade and cooperation.

**Keywords:** livestock industry; livestock products; livestock powerhouse; issues and challenges; policy recommendations

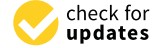



## 1. Introduction

The livestock industry is the pillar industry of China's agricultural and rural economy, and livestock products are important and indispensable foodstuffs for the population. Since reforming and opening up, China's livestock industry has made remarkable achievements in the development of livestock products, such as meat, eggs and milk, and the effective supply of such products to meet the population's needs. However, the livestock industry has long remained a fundamentally lagging aspect of agricultural and rural development. There are still, particularly in recent years, tight resource constraints, increased pressure on environmental protection, high external dependence on key factors, an international market that is somewhat weak in competition in addition to other outstanding issues [1–3]. Moreover, in the international livestock industry and livestock products market, there are challenges such as competition with large countries, trade friction, geopolitical risks, major natural disasters, animal and plant epidemics, public health emergencies and other instability and uncertainty factors [4–7]. Smooth and orderly development faces serious

challenges, and the resilience and security level of the livestock industry and supply chain need to be quickly improved.

From a global perspective, in 2021, the share of China's livestock production value in the total agricultural production value was maintained at 27.10%. Comparing major countries, the average share of livestock production value in the total agricultural production value is stable at approximately 41%, with Germany's share having exceeded 60% for a long time. China's meat, milk and egg production increased to 89.90 million, 37.78 million and 34.09 million tons, respectively, accounting for 25.39%, 4.49% and 36.83% of the world's total production. Among them, pork, beef, mutton and poultry production reached 52.96 million, 6.98 million, 5.14 million and 23.80 million tons, accounting for 44.00%, 9.63%, 31.43% and 17.25% of the world's total production, respectively. Although China has a large volume of domestic production, it still needs to import many livestock products. China's imports of pork, beef, mutton and poultry were 3.71 million, 2.33 million, 0.41 million and 1.34 million tons, respectively, and imports of dairy products were 4.93 million tons, all at a high level.

Based on the current development stage in the context of agricultural powerhouse construction, it is necessary to grasp and coordinate the shortcomings and milestones in the livestock industry's high-quality development, accelerate the construction of a livestock powerhouse as the main objective, achieve an all-round consolidation of the livestock industry and supply chain in terms of resilience and security, provide a firm material basis for Chinese-style modernization and build a strong socialist modern state. Therefore, it is of great practical significance to systematically evaluate China's livestock industry, determine the related issues and challenges, and study and propose policy recommendations to accelerate the construction of a livestock powerhouse.

There are currently no domestic or foreign studies on the construction of a livestock powerhouse. Related studies have focused more on exploring the competitiveness and high-quality development of China's livestock industry, only partially reflecting the realistic basis and strategic path for the construction of a livestock powerhouse in China. In terms of livestock industry competitiveness, the research analyzing the international competitive situation has addressed the perspectives of cost–benefit analysis, import and export trade, and production efficiency [8–10], and have measured international competitiveness using indicators such as international market share, trade competitiveness index, and revealed comparative advantage index [11–19]. They have also assessed international competitiveness by constructing evaluation index systems including multiple factors, such as resource endowment, production, consumption, quality, trade, and the environment [20–24]. Industry competitiveness and the influencing factors have been explored with the help of the diamond model, deviation-share analysis spatial model and structural equation model [25–28]. Many studies have considered the competitiveness of agriculture and livestock in other countries and regions, and their research methods and paradigms can be applied to China's case [29–38].

In terms of research findings, most studies conclude that China's livestock industry is not competitive. The main factors affecting the improvement in China's livestock industry competitiveness include market price, production cost, technology level, resource conditions, processing capacity, quality and safety, trade barriers, and policy support, and studies have suggested that multiple measures be taken to sustain efforts to fast-track the improvement in the international competitiveness of the livestock industry [39–42]. Other studies have considered the characteristics of the high-quality development of the industry [43–45] and systematically explored the challenges and practical paths for accelerating development [46–48].

Overall, current studies offer many insights regarding the competitiveness and high-quality development of China's livestock industry and support research and decision making regarding the construction of a livestock powerhouse. However, there is still room for exploration. On the one hand, current studies generally adopt indicators such as the international market share, trade competitiveness index and revealed comparative advantage

index, explore the competitiveness of China's livestock industry from the trade perspective and support the construction of a livestock powerhouse from a trade-competitiveness perspective. On the other hand, the evaluation index systems constructed to explore the competitiveness of the livestock industry include indicators such as cost and benefit, production efficiency and international trade, but do not consider the security of supply. Therefore, the assessment results do not reflect the reality of China's livestock industry competitiveness or the status of constructing a livestock powerhouse.

Considering that previous studies have focused more on the competitiveness and high-quality development of China's livestock industry, this study systematically discusses the strength of the livestock industry, conducts an international comparison, and outlines the issues, challenges and strategic choices from the perspective of supporting a strong agricultural industry. In 2022, the Chinese government clearly put forward that agricultural powerhouses should include "five strong" elements, namely, supply security, scientific and technological equipment, operation systems, industrial resilience and competitive ability. An index system for evaluating the strength of the livestock industry must be based on the current state of China's agriculture, reflect Chinese characteristics, and highlight important aspects, such as supply security and scientific and technological equipment. Therefore, this study established an evaluation index system based on the construction of a strong agricultural industry, evaluated the level of China's livestock industry development, compared it with the industries of other major countries, deeply explored the challenges and shortcomings, and finally put forward strategic paths and policy recommendations to support production and policy decisions.

The remainder of this paper proceeds as follows. Section 2 presents the methods and materials. Section 3 reports the results and discussion of the study. Section 4 describes the issues and challenges of China's livestock industry, and the final section presents the conclusions.

## 2. Methods and Materials

### 2.1. Methods

To build an evaluation index system for understanding the construction of a strong livestock industry, previous study findings, regarding supply security, scientific and technological equipment, industrial resilience, and competitiveness were analyzed based on the national livestock industry conditions of China.

### 2.1.1. Supply Security

Ensuring a stable and safe supply of grains and other important agricultural products is the top priority for accelerating the strengthening of agriculture in China. Likewise, resilient supply security is fundamental for the construction of a strong livestock industry. To enhance the supply of livestock products and ensure supply capabilities, the most critical and basic factor is the total amount supplied, which directly reflects the "strategic depth" and international competitiveness of the industry. In addition, the per capita occupancy and self-sufficiency are important, especially given China's large population and the need to support consumption demand, achieve autonomy, and provide nationally secure supplies. Ultimately, strong security is manifested when all 1.4 billion people have access to "meat, eggs and milk", which is a fundamental objective. Therefore, in the assessment of the strength of the livestock industry in terms of supply security, the three indicators of global share, per capita production and self-sufficiency rate of livestock products are key considerations.

### 2.1.2. Science and Technology Equipment

Science, technology and reform are important pillars in the construction of a strong agricultural industry. Science and technology are the first productive forces; with continuous advancements in the seed industry, machinery and other key areas of core technology and accelerating the transformation and application of scientific and technological achieve-

ments, we can improve the livestock industry's quality, efficiency, total factor productivity and competitiveness, and strengthen the foundation of scientific and technological support for further development. Therefore, to assess the level of scientific and technological equipment, the three indicators of livestock slaughter rate, slaughter carcass weight (yield) and agricultural labor productivity were mainly considered. These three indicators are the only ones considered because the number of patents, seed and machinery, contribution rate of scientific and technological progress, total factor productivity and other direct indicators, in addition to the meat-feed ratio and other indirect indicators that reflect the level of scientific and technological development in the field of livestock industry, were difficult to obtain. Notably, since the layer and dairy industries usually do not involve slaughter, only the two indicators of yield and agricultural labor productivity were considered for evaluation of these sectors.

### 2.1.3. Industrial Resilience

It is important to enhance the resilience and security level of the industry and supply chain. To accelerate the construction of a livestock powerhouse, we must continue to extend the industrial chain, enhance the value chain, stabilize the supply chain, strengthen the risk identification, control and transfer capabilities, and comprehensively ensure supply chain resilience and security. To assess the resilience of the livestock industry, this study mainly considered three indicators: the comparative benefits, resource carrying capacity and feed grain self-sufficiency rate. An industry's resilience can be evaluated according to whether its comparative benefits are high or low. We must rely on the market, policy and other channels to enhance the efficiency of livestock production and better coordinate local government and farm household efforts in agriculture and breeding. Moreover, given the continued tightening of resource and environmental constraints, the resource carrying capacity and feed grain supply should also be considered.

### 2.1.4. Competitiveness

In China, the government attaches great importance to improving agricultural quality, efficiency and competitiveness. To quickly strengthen the livestock industry, we must consider two markets and two sources of inputs, namely, the domestic and international markets. The international market and resources can support the domestic livestock product supply in multiple ways and channels, such as cooperation in international production capacity and global food and agriculture governance, enhanced rule-making power, product pricing power and resource control in the global arena and strengthened international competitiveness. Because competitive ability is more relevant in the field of international trade, this study considered the international market share, trade competitiveness index, and revealed comparative advantage index, as these were commonly used in the literature.

The four aspects mentioned above are also supply security, scientific and technological support, industrial resilience and international trade. The study considered the availability of relevant data and indictors for the livestock industry in various countries and relied on current research theories and methods in the construction of the evaluation framework. The specific indicators and calculation methods are shown in Table 1.

Considering the measurement differences across indexes with the aim to support comparison, this study first standardized the indicators, then calculated the weights of the indicators at all levels using the entropy method, and finally combined the standardized index values and weights to calculate the strength of the livestock industry. Because the indicators selected in this study are all positive indicators, their standardization formulae are as follows:

$$x_{ij} = \frac{a_{ij} - \min\{a_{ij}\}}{\max\{a_{ij}\} - \min\{a_{ij}\}} \tag{1}$$

where $x_{ij}$ is the standardized index value, $a_{ij}$ is the original value of the index, and $\max\{a_{ij}\}$ and $\min\{a_{ij}\}$ are the maximum and minimum values of the index, respectively.

**Table 1.** Evaluation index system of livestock powerhouse.

| | Indicators | Unit | Calculation Method |
|---|---|---|---|
| Supply security | Global share | % | Domestic livestock production/Global livestock production |
| | Per capita production | kg | Livestock production/Total population |
| | Self-sufficiency rate | % | Domestic livestock production/(Domestic livestock production + Net imports of livestock products) |
| Scientific and technological support | Slaughter rate | % | Livestock slaughter/Livestock inventory |
| | Carcass weight (yield) | kg/head | Livestock production/Livestock slaughter or Livestock production/Livestock inventory |
| | Agricultural labor productivity | USD/person | Value added in agriculture/Employment in agriculture |
| Industrial resilience | Comparative benefits | - | Producer price of livestock products/Producer price of feed grain |
| | Resource carrying capacity | Head/ha | Livestock inventory/Agricultural land |
| | Feed grain self-sufficiency rate | % | Domestic production of feed grain/(Domestic production of feed grain + Net imports of feed grain) |
| International trade | International market share | % | Domestic export value of livestock products/World export value of livestock products |
| | Trade competitiveness index | - | (Export value of livestock products − Import value of livestock products)/(Export value of livestock products + Import value of livestock products) |
| | Revealed comparative advantage index | - | (Domestic export value of livestock products/World export value of livestock products)/(Domestic export value of agricultural products/World export value of agricultural products) |

Before calculating the weights of each indicator through the entropy method, the weight of the corresponding indicator for the *i*th country under the *j*th indicator needs to be calculated as follows:

$$p_{ij} = \frac{x_{ij}}{\sum\limits_{i=1}^{n} x_{ij}} \tag{2}$$

where $p_{ij}$ is the corresponding indicator weight. Then, the entropy value of the *j*th indicator is calculated:

$$e_j = -k \sum_{i=1}^{n} p_{ij} \ln p_{ij} \tag{3}$$

where $e_j$ is the entropy value of the corresponding indicator and it satisfies $e_j \geq 0$; $k = 1/\ln n$. On this basis, the weights of each indicator are calculated as follows:

$$w_j = \frac{d_j}{\sum\limits_{j=1}^{m} d_j} \tag{4}$$

where $w_j$ is the corresponding indicator weight; $d_j$ is the information entropy redundancy of the *j*th indicator; and $d_j = 1 - e_j$.

*2.2. Materials*

Six categories of livestock species were considered in this study, including pig, beef cattle, sheep and goat, broiler, layer and dairy, and the basic data were all from the Food and Agriculture Organization of the United Nations (FAO) database. Except for the two indicators of agricultural value added and agricultural land area, which adopt 2020 data, the output of livestock products, import and export volume, import and export value and other indicators were all available in the 2021 data. After data processing, there were 112, 124, 108, 109, 108 and 132 countries considered for the pig, beef cattle, sheep and goats, broilers, layers and dairy industries, and 135 countries were considered for the whole

livestock industry. The samples selected in this study cover major countries such as China, the United States, Germany, France, New Zealand, Australia, Japan, Korea, Brazil, Russia and India, and the results of the evaluation are informative.

Because some of the larger countries with large production and export volumes of livestock products, such as Denmark, lacked price data or other key indicators, they were not considered in this study. In addition, most countries lacked detailed basic data on the output value of each livestock species and total livestock and agricultural employment, so the ratio of agricultural value added to the total population was used to measure agricultural labor productivity. Considering that feed grains were mainly corn and soybeans, and corn accounted for a higher proportion in the feed structure for livestock breeding, this study selected corn as representative for calculating the comparative benefits and feed grain self-sufficiency rate. Comparative benefits were measured by producer price of livestock products and producer price of corn, and the feed grain self-sufficiency rate was determined according to the corn self-sufficiency rate.

## 3. Results and Discussion

Based on the data regarding the livestock industry and related information for the evaluated countries, the indicators of supply security, scientific and technological support, industrial resilience and international trade were calculated after standardized processing, and the entropy method was used to obtain the strength of the livestock industry of the each country by species. The index weights were calculated according to the protein equivalent of livestock products using the standards of 15.1 g, 20.0 g, 18.5 g, 20.3 g, 13.1 g and 3.3 g of protein content per 100 g of pork, beef, mutton, poultry, eggs and milk, respectively, and the overall index was obtained according to the indexes of different species.

Figure 1 shows the index and ranking of China's livestock industry overall and by subspecies. In a comprehensive view, China's livestock powerhouse index was 0.2462, ranking 5th among the 135 countries considered. This indicates that China's livestock industry holds a relatively high position and is transforming from being merely a large-producing livestock country to being a livestock powerhouse. At the subspecies level, there are obvious differences in industry strength. China presents a strong layer industry, stronger hog, sheep and goat, and broiler industries, and a weaker beef cattle industry and dairy industry. China's powerhouse indexes for the pig, sheep and goat, and broiler industries are 0.3396, 0.2002 and 0.2730, respectively, ranking 6th, 5th and 7th among 112, 108 and 109 countries; this indicates that these industries are relatively strong in terms of worldwide comparison. The powerhouse indexes of the beef cattle industry and dairy industry are 0.1174 and 0.1011, ranking 34th and 42nd among 124 and 132 countries, respectively; this indicates that these industries are relatively weak. The powerhouse index of the layer industry is 0.5045, ranking 1st among 108 countries; this industry is strong and in the leading position in the world.

Table 2 provides the evaluation results for the top 20 countries, and Table 3 provides the state of livestock industry and subspecies industry performance for major countries. Regarding the overall livestock industry, the top five countries were the United States, Brazil, New Zealand, The Netherlands and China, with livestock powerhouse indexes of 0.4521, 0.3556, 0.3439, 0.2635 and 0.2462, respectively. Although China ranked high, there was still a very large gap compared with the United States, whose livestock powerhouse level was nearly two times that of China. There was also a large gap between China and Brazil and New Zealand. Of the other major countries, Australia and India ranked 9th and 10th, with livestock powerhouse indexes of 0.2075 and 0.1969, respectively. The top 20 countries included Germany, Canada and France, ranking 11th, 12th and 17th, respectively, with livestock powerhouse indexes of 0.1863, 0.1855 and 0.1608. Russia, Japan and Korea ranked relatively low, with ranks of 21st, 33rd and 37th and livestock powerhouse indexes of 0.1443, 0.1022 and 0.0957, respectively.

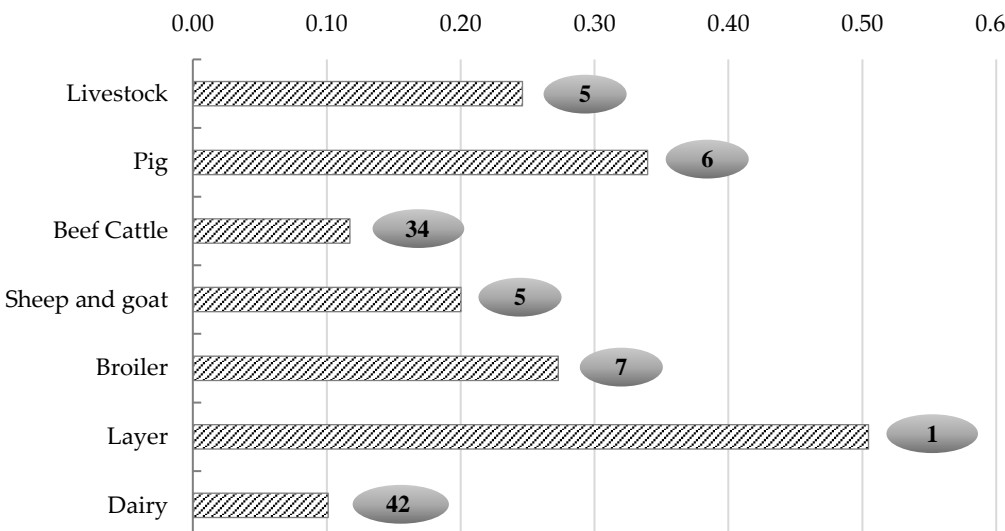

**Figure 1.** China's livestock powerhouse index and its ranking.

The top five countries in terms of the pig industry were Spain, the United States, The Netherlands, Germany and Canada. Spain's pig industry was obviously stronger than that of other countries. China ranked 6th after Canada, but there was still a gap compared to Spain, the United States, The Netherlands and other countries and a small gap compared to Canada and Germany. Of the other major countries, Brazil ranked 8th in the pig industry; France, Russia and South Korea ranked 14th, 15th and 18th, respectively; and India, Australia, Japan, New Zealand and other countries held lower positions (Tables 2 and 3).

The top five countries regarding the beef cattle industry were the United States, Brazil, Uruguay, Australia and New Zealand, and there was still a very large gap between China's beef cattle industry and those of the United States, Brazil and Australia. Other major countries, such as Canada, India, France, Germany, Japan, Russia and South Korea, had low ranks (Tables 2 and 3).

The top five countries for the sheep and goat industry were New Zealand, Australia, Ethiopia, Uruguay and China. Although China's sheep and goat industry ranked 5th in the world, there was a large gap compared to New Zealand and Australia. Of the other major countries, India ranked 9th in the world; France, Russia, Germany, Canada, the United States and Brazil ranked 29th, 40th, 46th, 55th, 63rd and 67th, respectively; and Japan and South Korea ranked relatively low (Tables 2 and 3).

The top five countries for the broiler industry were Brazil, the United States, Poland, Hungary and Belarus. Although China ranked 7th in the world, there was still a large gap compared with Brazil and the United States. Among other major countries, Australia, New Zealand, Russia, France and India were ranked in the world's top 20, and Canada, South Korea, Japan and other countries held relatively low positions (Tables 2 and 3).

The top five countries for the layer industry were China, the United States, Turkey, Panama and Malaysia, with China having the strongest layer industry, followed by the United States and Turkey. Of the other major countries, Germany, Japan, and India ranked 7th, 8th and 9th in the world, respectively, and Brazil, Canada, New Zealand, Russia, South Korea, Australia and other countries ranked relatively low (Tables 2 and 3).

The top five countries for the dairy industry were New Zealand, the United States, India, The Netherlands and Ireland. China ranked 42nd in the world, and there was a very large gap compared with New Zealand and the United States. Of the other major countries, Germany and France ranked 7th and 9th in the world, and Australia, Canada, Russia, Brazil, Japan and South Korea ranked relatively low (Tables 2 and 3).

**Table 2.** Evaluation results of livestock powerhouses.

| Livestock | | Pig | | Beef Cattle | | Sheep and Goat | | Broiler | | Layer | | Dairy | |
|---|---|---|---|---|---|---|---|---|---|---|---|---|---|
| Country | Index | Country | Index | Country | Index | Country | Index | Country | Index | Country | Index | Country | Index |
| United States | 0.4521 | Spain | 0.6049 | United States | 0.4727 | New Zealand | 0.7489 | Brazil | 0.6646 | China | 0.5045 | New Zealand | 0.7619 |
| Brazil | 0.3556 | United States | 0.4431 | Brazil | 0.4703 | Australia | 0.5163 | United States | 0.5592 | United States | 0.4680 | United States | 0.3678 |
| New Zealand | 0.3439 | The Netherlands | 0.4175 | Uruguay | 0.4320 | Ethiopia | 0.2048 | Poland | 0.4271 | Turkey | 0.4088 | India | 0.3210 |
| The Netherlands | 0.2635 | Germany | 0.3964 | Australia | 0.4031 | Uruguay | 0.2002 | Hungary | 0.3137 | Panama | 0.2841 | The Netherlands | 0.2938 |
| China | 0.2462 | Canada | 0.3430 | New Zealand | 0.3694 | China | 0.2002 | Belarus | 0.2998 | Malaysia | 0.2720 | Ireland | 0.2926 |
| Belarus | 0.2438 | China | 0.3396 | Paraguay | 0.3136 | Kenya | 0.1689 | The Netherlands | 0.2983 | Belarus | 0.2479 | Belarus | 0.2901 |
| Poland | 0.2338 | Belgium | 0.3331 | Ireland | 0.3096 | North Macedonia | 0.1686 | China | 0.2730 | Germany | 0.2475 | Germany | 0.2653 |
| Argentina | 0.2138 | Brazil | 0.2758 | Argentina | 0.3011 | Pakistan | 0.1633 | Brunei | 0.2678 | Japan | 0.2412 | Luxembourg | 0.2356 |
| Australia | 0.2075 | Austria | 0.2463 | Nicaragua | 0.2764 | India | 0.1495 | Turkey | 0.2647 | India | 0.2239 | France | 0.2342 |
| India | 0.1969 | Ireland | 0.2202 | Canada | 0.2341 | Moldova | 0.1429 | Ukraine | 0.2575 | Portugal | 0.2164 | Argentina | 0.2223 |
| Germany | 0.1863 | Hungary | 0.2143 | Brunei | 0.2139 | Spain | 0.1398 | Argentina | 0.2551 | Ukraine | 0.1929 | Uruguay | 0.2119 |
| Canada | 0.1855 | Italy | 0.2087 | The Netherlands | 0.2057 | Kazakhstan | 0.1377 | Thailand | 0.2439 | Brazil | 0.1833 | Belgium | 0.2102 |
| Turkey | 0.1855 | Chile | 0.1921 | Poland | 0.2047 | Greece | 0.1373 | Chile | 0.2236 | Brunei | 0.1832 | Poland | 0.2008 |
| Hungary | 0.1769 | France | 0.1913 | Belarus | 0.2028 | Kyrgyzstan | 0.1361 | Australia | 0.2057 | Argentina | 0.1829 | Australia | 0.1931 |
| Ukraine | 0.1690 | Russia | 0.1866 | Mexico | 0.1890 | Serbia | 0.1344 | New Zealand | 0.1901 | Morocco | 0.1825 | Czech Republic | 0.1855 |
| Spain | 0.1667 | Poland | 0.1851 | India | 0.1798 | Myanmar | 0.1290 | Russia | 0.1857 | Fiji | 0.1774 | Turkey | 0.1805 |
| France | 0.1608 | Belarus | 0.1591 | Pakistan | 0.1769 | Armenia | 0.1290 | Slovenia | 0.1852 | Thailand | 0.1744 | Kyrgyzstan | 0.1699 |
| Uruguay | 0.1602 | Korea | 0.1562 | Namibia | 0.1660 | Georgia | 0.1278 | France | 0.1612 | Pakistan | 0.1627 | Saudi Arabia | 0.1659 |
| Ireland | 0.1562 | Thailand | 0.1465 | Bolivia | 0.1535 | Argentina | 0.1182 | India | 0.1554 | Barbados | 0.1616 | Austria | 0.1649 |
| Paraguay | 0.1521 | Fiji | 0.1389 | South Africa | 0.1530 | South Africa | 0.1181 | Israel | 0.1516 | Bosnia and Herzegovina | 0.1512 | Slovenia | 0.1633 |

**Table 3.** Comparison of major countries' livestock powerhouse indexes.

| Country | Livestock | Pig | Beef Cattle | Sheep and Goat | Broiler | Layer | Dairy |
|---|---|---|---|---|---|---|---|
| Australia | 0.2075 | 0.0863 | 0.4031 | 0.5163 | 0.2057 | 0.1118 | 0.1931 |
| Brazil | 0.3556 | 0.2758 | 0.4703 | 0.0339 | 0.6646 | 0.1833 | 0.1199 |
| Germany | 0.1863 | 0.3964 | 0.1264 | 0.0480 | - | 0.2475 | 0.2653 |
| Russia | 0.1443 | 0.1866 | 0.0837 | 0.0575 | 0.1857 | 0.1271 | 0.1206 |
| France | 0.1608 | 0.1913 | 0.1455 | 0.0926 | 0.1612 | - | 0.2342 |
| Korea | 0.0957 | 0.1562 | 0.0651 | 0.0198 | 0.0870 | 0.1152 | 0.0786 |
| Canada | 0.1855 | 0.3430 | 0.2341 | 0.0402 | 0.1387 | 0.1484 | 0.1302 |
| United States | 0.4521 | 0.4431 | 0.4727 | 0.0368 | 0.5592 | 0.4680 | 0.3678 |
| Japan | 0.1022 | 0.0714 | 0.0974 | 0.0210 | 0.0834 | 0.2412 | 0.0906 |
| New Zealand | 0.3439 | 0.0589 | 0.3694 | 0.7489 | 0.1901 | 0.1461 | 0.7619 |
| India | 0.1969 | 0.0964 | 0.1798 | 0.1495 | 0.1554 | 0.2239 | 0.3210 |
| China | 0.2462 | 0.3396 | 0.1174 | 0.2002 | 0.2730 | 0.5045 | 0.1011 |

This study constructed an evaluation index system based on supply security, scientific and technological support, industrial resilience, and international trade to explore the overall situation of the livestock industry in the world, and the specific ranking was the result of comprehensive consideration of all four aspects. Given the state of China's livestock industry, its high ranking was mainly due to the volume of the livestock industry, with pork, beef, mutton and poultry production accounting for 44.00%, 9.63%, 31.43% and 17.25% of the world's total production, respectively, egg production accounting for 36.83%, dairy production accounting for 4.94%, and the global share of livestock products being generally high.

Naturally, different livestock species had different advantages in terms of supply security, scientific and technological support, industrial resilience, and international trade. For the layer industry, the scientific and technological support and industrial resilience indicators did not present advantages, while supply security and international trade did indicate advantages. China ranked first and third in the world in terms of the global share and per capita production of the layer industry, respectively; the self-sufficiency rate of eggs maintained a level higher than 100%; and the international market share and trade competitiveness index were fourth and first in the world, respectively. For the beef cattle industry and dairy industry, although the volume was large, the per capita production and self-sufficiency rate were at a low level, with the beef industry ranking 65th and 91st in the world and the dairy industry ranking 87th and 85th, respectively. The indicators reflecting the level of scientific and technological development, such as carcass weight and yield, were relatively weak. Due to the large net import volume, the international market share, the trade competitiveness index and the revealed comparative advantage index were relatively low.

Most previous studies concluded that China was not competitive in terms of sub-products of the hog, beef cattle, sheep and goat, broiler, layer, and dairy industries or the livestock industry as a whole [10,12–16,22,49]. This study, in contrast, found that China's livestock industry was generally strong, with the layer industry ranking first in the world, but the beef cattle and dairy industries were relatively weak. The main reason for the different findings is that this study constructed a more scientific evaluation index system, including supply security, scientific and technological support, industry resilience and international trade. Current studies have adopted the international trade perspective to explore the competitiveness and status of China's livestock industry.

## 4. Issues and Challenges of China's Livestock Powerhouse Construction

### 4.1. Supply Security Capacity Needs to Be Strengthened

With policy support and market-driven efforts, China's livestock industry has taken a new step in development, and the supply security capacity of livestock products has

significantly improved. However, there are still many weaknesses that need to be resolved. Although China maintains a high level in terms of total livestock products and ranks first in the world, its large population, high consumer demand and other factors yield a low per capita production and self-sufficiency rate. Since the livestock industry of the United States ranks first in the world, only the livestock industries in China and the United States will be compared here. Regarding per capita production, China's per capita production of pork, beef and poultry in 2021 were 37.14 kg, 4.89 kg and 16.69 kg, respectively, and these were less than 37.27 kg, 37.79 kg and 68.91 kg in the United States. The per capita production of dairy products in China was 25.83 kg, which was much lower than the value of 304.54 kg in the United States. In terms of self-sufficiency, China's self-sufficiency rates for pork, beef and poultry in 2021 were 93.39%, 74.94% and 95.02%, respectively, which were not as high as those of the United States, with levels of 120.75%, 100.38% and 119.28%. China's layer industry was highly competitive, but the self-sufficiency rate of 100.30% was slightly lower than that of the United States, 103.63%. China's dairy self-sufficiency rate was only 74.59%, far lower than that of the United States, which was 107.63% (Figure 2).

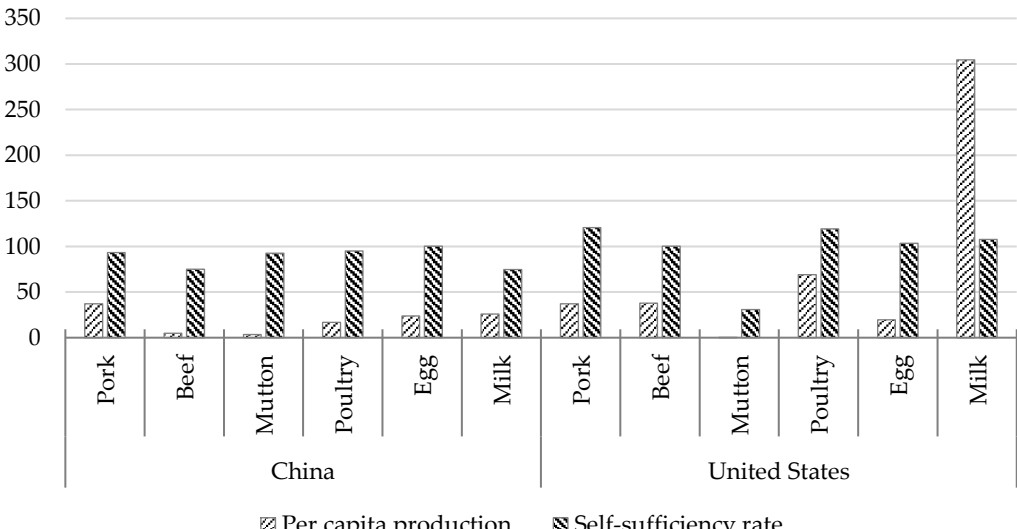

**Figure 2.** Comparison of the per capita production and self-sufficiency rate of livestock products between China and the United States in 2021 (kg, %). Data source: FAO database (https://www.fao.org/faostat/en/#home (accessed on 18 March 2023)).

The issues and challenges faced by livestock product supply security capacity enhancement were the result of the combined effects of the tightening resource and environmental constraints, insufficient scientific and technological support, weak operation systems, insufficient industry and supply chain resilience, increasing uncertainty and instability factors, and unsound policy support systems. Considering the systematic analysis of science and technology support, operation systems, industry and supply chain resilience, policy support systems, etc., only resource and environmental constraints will be discussed here. From the perspective of resource constraints, affected by the unequal exchange of factors between industry, agriculture and urban and rural areas and the constrained flow of such factors, land, labor and other resources were attracted to industry and cities. Agriculture and rural areas were less attractive to advantageous factors, and this was especially true for the development of the livestock industry.

In terms of land, the main manifestation was the difficulty in acquiring land for livestock breeding and the competition for land between rations and feed grain and forage. In terms of labor, the contradictions of poor quality and high cost were prominent, and the average annual growth of labor costs associated with the free-range breeding of pigs, beef cattle, sheep and goats in China from 2010 to 2021 were 7.19%, 9.33% and 9.46%, respectively. For the small-scale breeding of broilers and layers, the corresponding indicators

were 7.76% and 8.54%, respectively, and the average annual growth rate of labor costs in the free-range dairy cattle industry reached 8.82%. From the perspective of environmental protection, since 2015, the state has attached great importance to steadily promoting the green development of agriculture, with the successive introduction of "water 10" "soil 10" and other policy initiatives, and the green cycle of livestock industry development has advanced. However, environmental protection persistently represents the greatest challenge for livestock farming. In particular, regarding the implementation of environmental protection policies, some areas took unscientific and unreasonable "one-size-fits-all" measures to limit the development of the livestock industry, and farmers had to change their production approaches. The investment in environmental protection facilities and equipment directly increased the cost of livestock breeding and industry entry barriers.

*4.2. Scientific and Technological Support Needs to Be Improved*

Science and technology are the first productive forces, and innovation is a major driver of development. Thus, a strong livestock industry must rely on strong support from science and technology. Although China's scientific and technological innovation and applications in the livestock industry have achieved significant results, there are still many shortcomings reflected in the low production efficiency and high production costs. In the breeding industry, "foreign ternary" in the breeding pig market are becoming mainstream, and the quality of beef cattle breeds such as Simmental and Angus, major high-yielding dairy breeds such as Holstein, and sheep and goat breeds such as Suffolk, Boer and Dorper come from abroad [50]. Total mixed ration (TMR) mixers, automatic feeding carts, conveyor feeding systems, calf feeding carts, and other feeding machinery rely on imports with a proportion of 25%, robotic milking equipment with a proportion of 100%, other milking equipment with a proportion of 90%, environmental control machinery with a proportion of 70%, and manure treatment machinery with a proportion of 80% [51,52].

In terms of production efficiency in 2021, China's slaughter rate and carcass weight of pig, beef cattle, sheep and goat, and broiler were 133.56%, 77.98%, 110.99%, 252.04% and 88.27 kg/head, 148.19 kg/head, 14.49 kg/head and 1.55 kg/head, respectively, while layer and dairy cattle yields were 10.69 kg/head and 3026.90 kg/head, which were generally lower than those of the United States and other developed countries and lower than the world average (Table 4). Domestic livestock production costs continued to rise and were generally higher than the world average, resulting in an uncompetitive price advantage of domestic livestock products in the international market. At present, most of the livestock production cost growth rates are higher than the output value growth rate, and the total costs of production of pigs, sheep and goats, broilers and layers from 2010 to 2021 increased by 7.75%, 7.98%, 3.73% and 3.37%, respectively, and these rates were higher than the corresponding 6.62%, 7.37%, 2.35% and 2.89% of the total output value. Although the total production costs of beef cattle and dairy cattle grew at a slightly lower rate than the total output value, they were still relatively high, with average annual growth rates of 10.18% and 4.30%, respectively (Table 5).

Comparing the production costs of pigs in China and the United States, the production costs of pigs in the United States in 2021 were only 51.96%, 57.46%, 59.05% and 60.25% of the corresponding levels of free-range, small-scale, medium-scale and large-scale in China, respectively. Similar conclusions can be drawn from the import prices of livestock products and domestic market prices. In 2021, the average import prices of pork, beef and mutton were 2.62 USD/kg, 5.35 USD/kg and 5.79 USD/kg, respectively, and the domestic market prices in that year were 5.03 USD/kg, 13.20 USD/kg and 13.04 USD/kg, respectively. The average import price of poultry was 2.37 USD/kg, and the average import price of dairy was 0.57 USD/kg, both lower than the domestic levels of 3.40 USD/kg and 0.60 USD/kg above. In general, China's scientific and technological innovation in the livestock industry needs to be strengthened, and there is still much room for improvement in the construction of a livestock industry empowered by science and technology.

**Table 4.** Comparison of livestock production efficiency in major countries in 2021 (%, kg/head).

| Country | Pig | | Beef Cattle | | Sheep and Goat | | Broiler | | Layer Yield | Milk Yield |
|---|---|---|---|---|---|---|---|---|---|---|
| | Slaughter Rate | Carcass Weight | Slaughter Rate | Carcass Weight | Slaughter Rate | Carcass Weight | Slaughter Rate | Carcass Weight | | |
| Australia | 212.97 | 78.74 | 27.10 | 291.88 | 37.87 | 24.69 | 608.94 | 1.92 | 16.01 | 6400.40 |
| Brazil | 124.70 | 82.29 | 12.34 | 351.92 | 30.09 | 14.46 | 400.96 | 2.43 | 12.30 | 2280.70 |
| Canada | 166.99 | 102.63 | 30.01 | 417.46 | 94.06 | 22.99 | 439.12 | 1.89 | 17.72 | 9646.70 |
| China | 133.56 | 88.27 | 77.98 | 148.19 | 110.99 | 14.49 | 252.04 | 1.55 | 10.69 | 3026.90 |
| France | 180.15 | 94.53 | 25.73 | 319.48 | 57.87 | 18.14 | 305.50 | 1.89 | - | 7458.90 |
| Germany | 218.31 | 95.83 | 29.60 | 330.61 | 99.05 | 20.41 | - | - | 19.70 | 8481.40 |
| India | 103.64 | 35.00 | 21.08 | 103.00 | 34.88 | 10.59 | 328.87 | 1.34 | 11.98 | 1880.20 |
| Japan | 181.24 | 78.29 | 26.82 | 450.00 | 30.31 | 26.20 | 261.72 | 2.88 | 18.30 | 8939.20 |
| New Zealand | 254.74 | 70.89 | 46.27 | 159.81 | 87.20 | 20.24 | 458.88 | 1.91 | 15.96 | 4555.30 |
| Korea | 163.89 | 76.54 | 24.18 | 321.24 | 39.24 | 15.45 | 583.80 | 0.94 | 10.09 | 10,374.30 |
| Russia | 179.23 | 92.90 | 43.43 | 213.75 | 55.13 | 18.04 | 484.88 | 1.89 | 16.70 | 5016.70 |
| United States | 174.06 | 97.32 | 36.64 | 370.59 | 38.08 | 24.92 | 597.13 | 2.43 | 17.08 | 10,869.00 |

Data source: FAO database (https://www.fao.org/faostat/en/#home (accessed on 18 March 2023)).

**Table 5.** Trends in the cost–benefit analysis of China's livestock industry.

| Type | | 2010 | 2015 | 2020 | 2021 | Average Annual Growth Rate (%) |
|---|---|---|---|---|---|---|
| Pig (USD/head) | Total output | 198.10 | 293.36 | 601.12 | 400.91 | 6.62 |
| | Total cost | 184.68 | 294.67 | 422.42 | 419.93 | 7.75 |
| | Feed cost | 102.95 | 139.70 | 139.26 | 172.47 | 4.80 |
| | Labor cost | 35.30 | 82.01 | 78.00 | 79.49 | 7.66 |
| Beef cattle (USD/head) | Total output | 888.53 | 1712.04 | 2423.05 | 2712.35 | 10.68 |
| | Total cost | 736.15 | 1372.85 | 1832.71 | 2138.93 | 10.18 |
| | Feed cost | 190.73 | 274.48 | 366.36 | 459.66 | 8.33 |
| | Labor cost | 65.09 | 162.24 | 166.37 | 182.21 | 9.81 |
| Sheep and goat (USD/head) | Total output | 114.54 | 150.29 | 228.14 | 250.31 | 7.37 |
| | Total cost | 94.49 | 160.89 | 198.31 | 219.83 | 7.98 |
| | Feed cost | 24.23 | 32.41 | 35.08 | 41.89 | 5.10 |
| | Labor cost | 28.99 | 67.81 | 71.77 | 82.24 | 9.94 |
| Broiler (USD/hundred head) | Total output | 367.75 | 421.79 | 399.23 | 474.59 | 2.35 |
| | Total cost | 327.87 | 433.55 | 420.84 | 490.43 | 3.73 |
| | Feed cost | 238.70 | 306.96 | 277.84 | 340.18 | 3.27 |
| | Labor cost | 26.56 | 56.87 | 60.04 | 63.43 | 8.23 |
| Layer (USD/hundred head) | Total output | 2034.49 | 2507.39 | 2026.89 | 2783.14 | 2.89 |
| | Total cost | 1901.46 | 2351.13 | 2300.09 | 2736.70 | 3.37 |
| | Feed cost | 1411.96 | 1618.85 | 1555.95 | 1924.03 | 2.85 |
| | Labor cost | 108.12 | 256.83 | 269.05 | 279.43 | 9.02 |
| Dairy cattle (USD/head) | Total output | 2416.97 | 3425.87 | 3567.85 | 4063.75 | 4.84 |
| | Total cost | 1800.86 | 2622.05 | 2508.15 | 2861.66 | 4.30 |
| | Feed cost | 1258.11 | 1632.09 | 1487.27 | 1755.73 | 3.08 |
| | Labor cost | 244.01 | 593.31 | 608.94 | 648.91 | 9.30 |

Note: Data from the National Compilation of Cost and Benefit Information of Agricultural Products. The costs and benefits of pig, beef cattle, sheep and goat, and dairy cattle are for free-range breeding, and the costs and benefits of broiler and layer are for small-scale breeding.

*4.3. The Modern Operation System Needs to Be Sound*

China's government has attached great importance to the construction of the agricultural industry system and its production and operation systems. A series of policies and measures have been formulated to accelerate the construction of a new operation system, promote the continuous growth of new business entities in the livestock industry, promote the continuous growth of industrialized leading enterprises, enhance professional cooperatives and family farms, and accelerate the development of socialized service organizations.

However, compared with developed countries, China still needs to transform and upgrade its livestock operation system. In terms of scale, in 2021, China's livestock breeding scale rate reached 69.0%, an increase of 48.4 percentage points compared with 2003. Among these rates, the scale rates of pig, beef cattle, sheep and goat, broiler, layer and dairy breeding reached 62.0%, 32.9%, 44.7%, 85.7%, 81.9% and 70.8%, respectively, and the scale of major livestock breeding greatly improved. Except in the cases of beef cattle and sheep and goats, the scale of other major livestock breeding was at a high level (Figure 3).

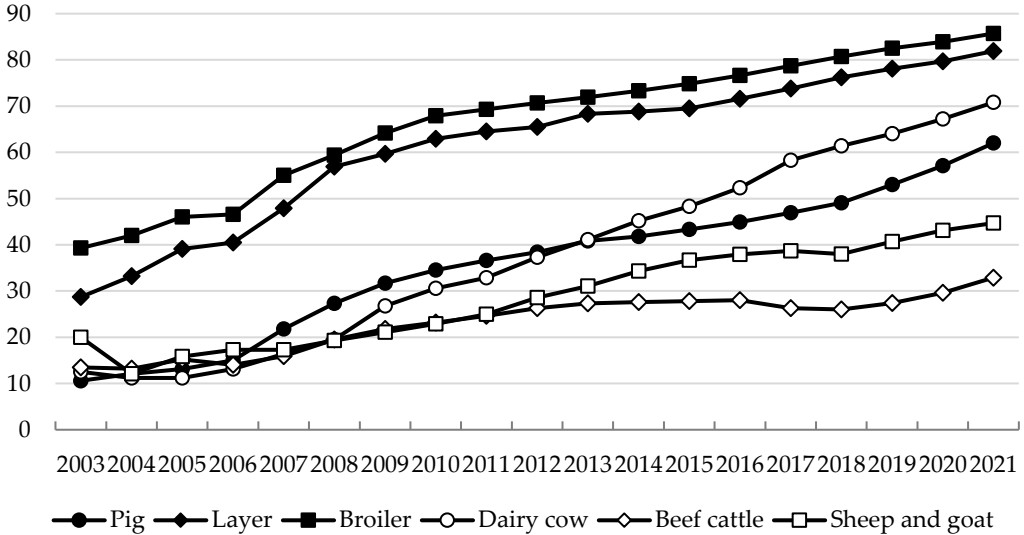

**Figure 3.** China's livestock breeding scale rate trend (%). Note: Data from China Animal Husbandry and Veterinary Statistics. The different livestock breeding scale standards are as follows: more than 500 pigs slaughtered per year, more than 2000 layers inventory per year, more than 10,000 broilers slaughtered per year, more than 100 dairy cows inventory per year, more than 50 beef cattle slaughtered per year, and more than 100 sheep and goats slaughtered per year.

In terms of industry concentration, although the concentration of China's livestock industry has increased significantly, it is still at a relatively low level. Taking pig breeding as an example, after the African swine fever outbreak in 2018, there was a new round of "reshuffling" in the pig industry, with the total number of pigs slaughtered by the top 10 listed pig enterprises in China increasing from 49.55 million to 124.5 million from 2018 to 2022 (subject to the ranking in 2022), an increase of 151.26%, and the share of national slaughter increasing from 7.14% to 17.79%, an increase of 10.65 percentage points (Table 6). The industry concentration of pig breeding has rapidly improved. The concentration of the pig slaughtering industry has gradually increased, and the slaughter volume of the above-scale pig slaughtering enterprises nationwide increased significantly in 2022, with a slaughter volume of 285.37 million head, an increase of 17.67% compared with 2018. The slaughter rate of fixed location slaughtering grew steadily, from 34.95% in 2018 to 40.77% in 2022, an increase of nearly 6 percentage points. China Business Intelligence research data show that the CR5 (five-firm concentration ratio) share of China's slaughter industry was approximately 5%, of which the slaughter leader's share was only 2.35%.

Compared to the United States and other developed countries, China's livestock industry presented a low scale and industry concentration, and there was more room for future improvement. According to United States Department of Agriculture (USDA) data, in 2012, the proportion of pig inventory above 500 head in the total inventory in the United States reached 97.3%, and the scale level was much higher than the corresponding level in China [53]. The concentration of pork processing was also at a high level, with the market share of the top four slaughter and processing enterprises reaching approximately 70%.

**Table 6.** Trend of pig slaughter of listed pig enterprises in China, 2018–2022 (million head, %).

| Enterprise | 2018 | 2019 | 2020 | 2021 | 2022 |
|---|---|---|---|---|---|
| Muyuan Foods Co., Ltd. | 11.01 | 10.25 | 18.12 | 40.26 | 61.20 |
| Wens Foodstuff Group Co., Ltd. | 22.30 | 18.52 | 9.55 | 13.22 | 17.91 |
| New Hope Group | 2.55 | 3.55 | 8.29 | 9.98 | 14.61 |
| Zhengbang Group | 5.54 | 5.78 | 9.55 | 14.93 | 8.45 |
| Aonong Group | 0.42 | 0.66 | 1.35 | 3.25 | 5.19 |
| Da Bei Nong Group | 1.68 | 1.64 | 1.85 | 4.31 | 4.43 |
| Tech-bank Food Co., Ltd. | 2.17 | 2.44 | 3.08 | 4.28 | 4.42 |
| COFCO Joycome Foods Limited | 2.55 | 1.99 | 2.05 | 3.43 | 4.10 |
| Tangrenshen Group Co., Ltd. | 0.68 | 0.84 | 1.02 | 1.54 | 2.16 |
| Tecon Biology Co., Ltd. | 0.65 | 0.84 | 1.35 | 1.60 | 2.03 |
| Total | 49.55 | 46.51 | 56.21 | 96.80 | 124.50 |
| National | 693.82 | 544.19 | 527.04 | 671.28 | 699.95 |
| Percentage of | 7.14 | 8.55 | 10.67 | 14.42 | 17.79 |

Note: Data from the National Bureau of Statistics and related enterprises. This table was sorted according to the top 10 listed pig enterprises in terms of slaughter volume in 2022.

*4.4. The Industry and Supply Chain Are Not Highly Resilient*

Since the reform and opening-up, China's livestock industry policy system has been improved to promote the resilience and security of the livestock industry and supply chain, gradually improving the effective response to "black swan" and "gray rhino" event impacts. However, the major task of extending, complementing and strengthening the supply chain of the livestock industry remains; the industry business model still needs innovation; the level of industrial integration needs to be enhanced; risk control and transfer capacity need to be strengthened; and the livestock industry and supply chain resilience and security need to be firmly established.

A traditional Chinese saying states that "Family money with animal does not count", referring to the major risk of animal disease for livestock breeding. For example, comparing the situation of major animal diseases in China and the United States since 2010, the epidemic situation in China has been more severe. This situation reflects the weak level of domestic-animal disease prevention and control measures and the need for efficient prevention and control of epidemics to help build a powerful livestock industry. In 2010–2016, there were more than 20,000 new outbreaks of animal diseases in China. In 2017–2019, the new outbreaks of animal diseases were more than 1000. However, since 2020, with the improvement in animal disease prevention and control systems and the strengthening of prevention and control capacity, new outbreaks have been significantly reduced. In contrast, in the United States, new outbreaks of animal diseases have never exceeded 1000 since 2010, showing the advantages of animal disease prevention and control in the United States (Table 7).

Considering the feed grain and forage supply security capacity, corn, soybeans and other feed materials in China are highly dependent on imports. Grain-based livestock, including the layer, pig and broiler industries, present greater advantages than the dairy and beef cattle industries, but the risk of a break in the supply chain remains. If feed grain cannot be imported, a situation of extreme emergency is likely to result, such that China's livestock industry cannot be considered very strong. In addition, after the "melamine" incident in 2008, farmers paid more attention to the role of high-quality forage in the diet of grass-fed livestock, and the ratio of green and roughage feed to concentrate feed for grass-fed livestock showed an overall increase. However, the domestic forage industry started late and lagged in development. In 2020, nearly 80 million mu of high-quality forage was planted on arable land, and the forage output reached 71.6 million tons, of which 6.5 million mu was planted with high-quality and high-yield alfalfa, and the output was 3.4 million tons. In contrast, in the United States, alfalfa has become the fourth largest crop after corn, wheat, and soybeans, and its production volume in recent years has been maintained at more than 11 million tons, making it an important driver of dairy production

efficiency and competitiveness. Currently, China's dairy cattle inventory is 11 million head, and its milk production is 39.3 million tons. Vigorously developing a high-quality forage industry is important to improve the quality and efficiency of milk production, enhance supply chain resilience and strengthen the competitiveness of China's dairy industry.

**Table 7.** Comparison of the state of major animal diseases in China and the United States.

| Year | China | | | | United States | | | |
|------|-----------------|------------------------|---------------------------------------|-----------------------------|-----------------|------------------------|---------------------------------------|-----------------------------|
| | New Outbreaks | Cases (Thousand Head) | Killed and Disposed (Thousand Head) | Deaths (Thousand Head) | New Outbreaks | Cases (Thousand Head) | Killed and Disposed (Thousand Head) | Deaths (Thousand Head) |
| 2010 | 20,434 | 4965.62 | 183.62 | 700.79 | 53 | 0.73 | 1.35 | 0.04 |
| 2011 | 30,466 | 4179.31 | 313.50 | 435.63 | 40 | 0.31 | 15.47 | 0.00 |
| 2012 | 24,107 | 2690.81 | 1844.96 | 320.46 | 50 | 0.92 | 0.17 | 0.00 |
| 2013 | 24,884 | 2670.37 | 611.74 | 332.65 | 55 | 6.81 | 10.65 | 0.00 |
| 2014 | 22,640 | 2652.07 | 2171.67 | 225.36 | 454 | 0.75 | 116.49 | 0.07 |
| 2015 | 27,142 | 2933.46 | 623.42 | 349.67 | 253 | 25.27 | 16,071.44 | 59.23 |
| 2016 | 23,266 | 2816.38 | 1002.05 | 487.75 | 25 | 0.68 | 236.72 | 1.04 |
| 2017 | 8381 | 1008.25 | 772.34 | 289.63 | 25 | 0.59 | 195.92 | 0.70 |
| 2018 | 8560 | 859.06 | 636.46 | 205.18 | 35 | 0.57 | 221.95 | 0.21 |
| 2019 | 3946 | 318.38 | 202.71 | 52.97 | 52 | 0.92 | 1714.60 | 0.16 |
| 2020 | 124 | 9.02 | 17.59 | 8.73 | 245 | 0.85 | 339.47 | 22.16 |
| 2021 | 120 | 5.92 | 4.34 | 5.79 | 82 | 0.37 | 0.39 | 0.26 |
| 2022 | 18 | 0.31 | 0.60 | 0.29 | 994 | 0.17 | 56,914.34 | 1444.38 |

Data source: World Organization for Animal Health (https://wahis.woah.org/ (accessed on 18 March 2023)).

*4.5. International Trade Risks Were Increasing*

In recent years, affected by the tightening of domestic resources and environmental constraints, the expansion of urban and rural residents' consumption needs and other factors, China's livestock industry has continued to increase the depth and breadth of its opening to the world. China has also increased its use of international markets and resources to compensate for the lack of domestic resource endowment and the gap between the supply and demand of livestock products. In particular, in the context of China's grain security problem, manifested mainly in the security of feed grain, livestock products and feed grain imports to save domestic resources, corn, soybeans and other feed grain imports were equivalent to the use of nearly 800 million acres of foreign arable land. However, the current use of international markets and resources to support the high-quality development of the domestic livestock industry and the livestock industry entails increasing risks.

On the one hand, the risk of a high concentration of imports of livestock products and feed grains should pose a high alert. From the perspective of import source countries, pork imports have come mainly from Spain, Brazil and Denmark; beef imports mainly from Brazil, Argentina and Uruguay; mutton imports from New Zealand and Australia; poultry imports from Brazil and the United States; and dairy imports mainly from New Zealand and the European Union. In 2022, 92% of soybean imports were from the United States and Brazil, of which Brazil accounted for nearly 60%; 98% of corn imports were from the United States and Ukraine, with the United States accounting for 72%; and over 60% of sorghum imports were from the United States.

On the other hand, livestock products and feed grain imports face the risk of growing uncertainty. In a century of unprecedented changes, the game, unilateralism, protectionism, hegemony, etc., continue to intensify, as the United States and other developed countries seek to contain China, and these changes are still unabated. Trade disputes and scientific and technological suppression continue to exist; in extreme cases that do not exclude feed grain, livestock products face the same pressure as the "chip" market. The conflict between Russia and Ukraine is ongoing, and the geopolitical risks regarding the international supply of livestock products and feed grain cannot be ignored (Figure 4). In the international

competition in the livestock and related industries between China and the United States, the United States initiative has been stronger; and its advantage more obvious. In addition, due to the frequency of extreme weather disasters worldwide, with major animal and plant epidemics, public health emergencies and other disasters, it is difficult to provide a stable environment for the safe import of livestock products, feed grains, etc.

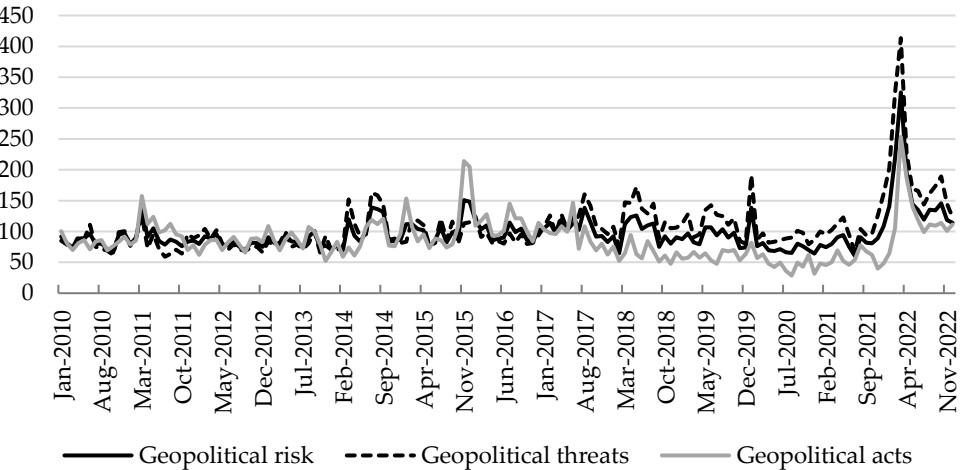

**Figure 4.** Global geopolitical risk trends, 2010–2022. Data source: Economic Policy Uncertainty Database (http://www.policyuncertainty.com/index.html (accessed on 18 March 2023)).

*4.6. The Policy Support System Needs to Be Improved*

The policy support system of China's livestock industry has gradually improved. This is driven by many factors, such as changes in consumer demand, tightening resource and environmental constraints, and increasing external uncertainties. Currently, a policy support system has formed, covering all aspects of the whole industry chain, such as breeding, slaughtering and processing, circulation, storage, disease prevention and control, green development and domestic and international market resources. This policy system has played an important role in strengthening the supply security capacity of livestock products, improving the quality, efficiency and competitiveness of the livestock industry and accelerating the construction of a livestock powerhouse. However, compared with countries and regions such as the United States and the European Union, China's livestock policy support system can be further improved and optimized, especially in terms of policy foresight, precision and sustainability.

Taking African swine fever as an example, after it was first diagnosed in Liaoning Province in August 2018, the epidemic spread rapidly across the country within a short period. This had a negative impact on the pig industry and market and seriously damaged the pig production base, highlighting the contradiction between supply and demand in the pork market and revealing the high cost and difficulty of providing meat for the population [54,55]. Reflecting on the current round of epidemic prevention and control measures and pig market regulation policies, the government has intensively introduced a series of policy initiatives in the short term, including the release of important policy documents such as the "Opinions of the General Office of the State Council on Strengthening Prevention and Control of African Swine Fever", "Three-Year Action Plan for Accelerating the Recovery and Development of Pig Production", and "Opinions of the General Office of the State Council on Promoting the High-Quality Development of the Livestock Industry". However, the control policies implemented in response to African swine fever do not correspond appropriately to the situation but rather deepen market fluctuations and compress the original "pig cycle" [56,57].

On the one hand, even though major animal epidemics, such as porcine reproductive and respiratory syndrome and highly pathogenic avian influenza, were experienced in the early stage, the lessons were not well absorbed to effectively address African swine

fever, resulting in the wide spread of the epidemic and large-scale losses in pig breeding. On the other hand, pig market control policies followed the epidemic, showing a certain degree of passivity and lagging, with policy initiative and foresight still to be strengthened (Figure 5). The policy initiatives related to the current round of the epidemic did not achieve better results in suppressing the sharp rise and rapid fall of pig market prices, and many farmers suffered a "second blow" due to the drastic market shock. Overall, China's livestock industry policy support system still needs to be improved to strengthen the policy assurance and institutional governance for the construction of a livestock powerhouse.

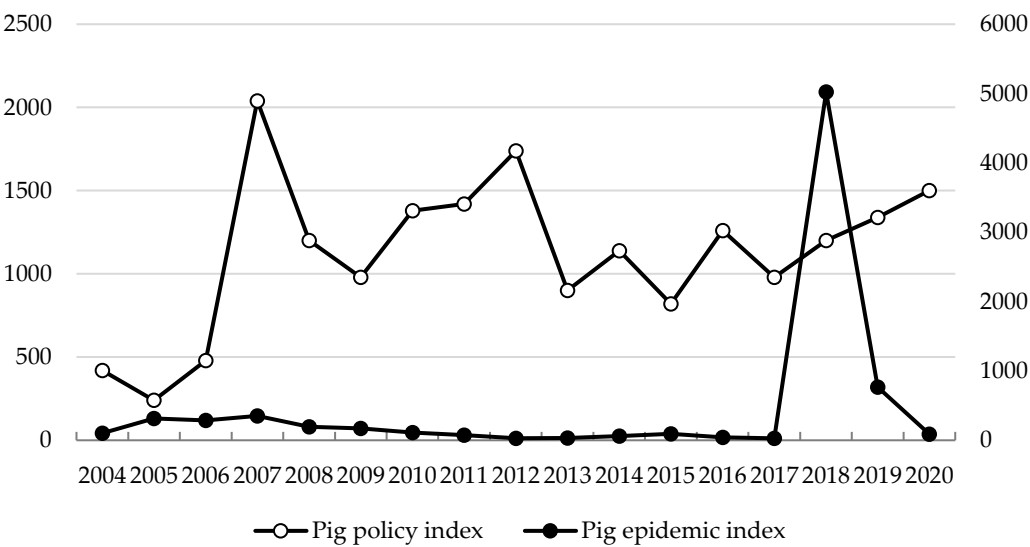

**Figure 5.** Trend of China Pig Epidemic and Policy Index. Note: Data from [57,58].

## 5. Conclusions

This study has constructed an evaluation index system using data related to the livestock industry for major countries in the world, used the entropy method to comparatively assess the strength of China's livestock industry and discussed the challenges and shortcomings in constructing a livestock powerhouse. Specifically, the following research conclusions can be obtained.

First, China's livestock industry ranked high, indicating that China has been moving from a large livestock-producing country to a live-stock powerhouse. China's livestock industry ranked 5th overall among 135 countries. Although the ranking was relatively high and the strength was relatively robust, there was still a large gap compared to the United States and compared with the domestic livestock aims and needs, indicating the need for further advancement. China's livestock industry ranked higher than the industries of Australia, Germany, Canada, France, Japan, South Korea and other countries. There were obvious differences across different livestock species. China's layer industry ranked 1st out of 108 countries. The pig, sheep and goat, and broiler industries were relatively strong, ranking 6th, 5th and 7th out of 112, 108 and 109 countries, respectively. The beef cattle and dairy industries were weak, ranking 34th and 42nd out of 124 and 132 countries, respectively.

Second, the construction of a strong livestock industry still faces many issues and challenges. Domestic supply security capacity needs to be strengthened, and international trade risks are increasing. The high cost and low efficiency of livestock production problems are evident, and the level of scientific and technological support needs to be improved. The livestock production scale needs to be intensified, and the modern operation system needs to be sound. The animal disease prevention and control capacity is weak, instability and uncertainty have increased, and the industry and supply chain resilience is not strong. The policy accuracy and effectiveness are not sufficient, and the policy support system needs to be improved.

Based on the above findings, the following strategic path and policy recommendations are proposed for reference. First, improve the institutional mechanism for the construction of a livestock powerhouse. Develop a strong livestock-industry development program and implement strong measures; adhere to the national, unified leadership and planning layout; and clarify the responsibilities and obligations of various departments and different production and business entities. Focus on supply security, science and technology equipment, business systems, industrial resilience, competitiveness and other key tasks; further improve the policy support system; increase the financial and financial insurance support breadth and depth; and leverage social forces to actively invest in the construction of a strong livestock industry.

Second, promote a high level of self-reliance and self-improvement in livestock-industry science and technology. Focusing on the seed industry, machinery and other key areas of core technology, we should increase the investment in livestock-industry science and technology; strengthen the strategic layout of livestock-industry science and technology; deepen the basic, frontier, public welfare and strategic research; and support the innovative transformation of scientific and technological achievements and technology promotion. The soybean-oil production capacity improvement project should be deepened, and the corn yield improvement project should be implemented to develop high-quality feed grain and strengthen the feed forage domestic supply security.

Third, a modern-livestock-industry business system needs to be constructed. We should strengthen the cultivation of new business entities and further support leading enterprises, professional cooperatives, family farms and other aspects of livestock industry, accounting for the development of traditional small farmers and retail livestock and poultry breeding. We should adopt innovative "enterprise + family farm", "enterprise + cooperative", "enterprise + farmers" and other business models, improve the close interest linkage mechanism, give full play to the leading role of leading enterprises, and help small farmers adopt modern methods of livestock industry.

Fourth, enhance the industry supply-chain resilience and security level. Market monitoring and early warning should be strengthened, a monitoring and early warning system with global influence should be built, short-term and medium- and long-term reports should be regularly released, market assessments should be made, and contingency plans should be clarified. We should promote the open socialization of market information data so that policy makers, producers, consumers, researchers and others can access useful information. We should accelerate the cultivation of new industries and new models in the livestock industry to further extend the industrial chain, enhance the value chain and stabilize the supply chain. We should innovate livestock insurance systems, strengthen the construction of animal disease prevention and control systems, and improve the policy toolbox for responding to unexpected and uncertain events.

Fifth, international trade and cooperation should be consolidated and expanded. According to the strength of the competitive ability, we should adjust import and export priority strategies by species. We should continue to strengthen the existing international cooperation base and actively expand new trade channels. We should cultivate livestock enterprises with global influence to support the export of advantaged livestock and poultry industries in a gradual manner, and enhance the ability to control the supply chain of the global livestock industry chain.

This study offers a comparative analysis of China's livestock industry by constructing an evaluation index system. However, because some indicators and data are not available, many countries are not in the scope of evaluation, and many indicators are not included in the evaluation index system. In particular, this study first considered the selection of indicators related to the operating system. The development situation of enterprises, cooperatives, family farms, socialized service organizations and other business subjects and their indicators, such as scale and intensification, should be considered, but it was impossible to obtain large-scale data from various countries to support a comparative analysis. Therefore, this study did not consider indicators related to the operation system

when evaluating the livestock industry but systematically discussed them in Section 4 of this article. In the future, the possibility of constructing an evaluation index system with more indicators can be considered to conduct a more systematic and in-depth evaluation of important developed countries involved in the livestock industry.

**Author Contributions:** Conceptualization, Z.S., J.L. and X.H.; methodology, Z.S., J.L. and X.H.; resources, Z.S., J.L. and X.H.; data curation, Z.S. and J.L.; writing—original draft preparation, Z.S., J.L. and X.H.; writing—review and editing, Z.S., J.L. and X.H.; visualization, Z.S., J.L. and X.H.; supervision, Z.S. and X.H.; project administration, Z.S. and X.H.; funding acquisition, Z.S. and X.H. All authors have read and agreed to the published version of the manuscript.

**Funding:** This research was funded by the National Social Science Fund of China (22CGL025), National Natural Science Foundation of China (72033009), Central Public-interest Scientific Institution Basal Research Funds (1610052023013), and Agricultural Science and Technology Innovation Program (10-IAED-01-2023).

**Institutional Review Board Statement:** Not applicable.

**Data Availability Statement:** Data available on request from the authors.

**Conflicts of Interest:** The authors declare no conflict of interest.

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
