# Peer review of "From Large to Powerful: International Comparison, Challenges and Strategic Choices for China’s Livestock Industry"

_agriculture, doi:10.3390/agriculture13071298_

Round 1

Reviewer 1 Report

Dear Editor and Authors,

I hope to contribute with improvements and advances in relation to your study. Questions and suggestions follow.

In the introduction, the authors develop the context of the theme with only 4 long paragraphs. I recommend a reorganization:

In lines 33 to 57 – there is a lack of authors to justify the statements presented, as well as the authors could initially speak of the world context, and later address the specific context/problems of China. Also because the analysis is developed in a comparative way.

In lines 58 to 82 – it is necessary to separate ideas and general studies from those that are related to your analysis. What do the studies show? They are broad and distinct subjects, what do you want to show the reader? It's too confusing for the search input.

 In lines 105 and 112 - “Therefore, this study established an evaluation index system based on the construction of agricultural powerhouses, evaluated the level of China’s livestock powerhouse, compared it with major countries in the world, deeply explored the challenges and shortcomings faced by the construction of livestock powerhouses, and finally put forward strategic paths and policy recommendations to be referenced in production and policy decisions”.

I would like the authors to reflect, does this really happen? Is the analysis developed from production data already available, how does the comparison add differentials in the literature, what are the specificities of China in relation to other countries? Because the 4 dimensions of analysis were not discussed in the introduction, I suggest that there is this focus.

In Methodology, page 132 to 219 – the authors describe in items 2.1.1, 2.1.2, 2.1.3, 2.1.4 and 2.1.5 – aspects of the study environment at country level and make reference to the party document Chinese Communist. However, as a reader I could not identify:

- what methodological steps were followed?

- what is the route for data collection, data processing? And proposed form of analysis?

- In Table 1, where did the indicators used come from? Who defined the unit and method? Who validated? What is the time period? The analysis of these indicators will be developed from which data? What's the source?

 Note that section 2.2 should have been presented earlier, so that the reader could understand how data collection and processing was developed.

I suggest to the authors improvement in the presentation of the methodology, unifying sections 2.1 and 2.2. Subsequently subsections 2.1.1, 2.1.2, 2.1.3, 2.1.4 and 2.1.5 – be reformulated, it may be Table 1 later, commenting on what the 4 dimensions presented in Table 1 represent, later they will be explained in the analysis, not the method, ok!

Note that the sources used in the analysis, in the graphs and tables – were not cited in the methodology.

The biggest deficiency of the study is the methodological alignment, it lacks to clarify the way, the stages of the development of the research to the readers.

When discussing the results, the authors should use the same basis as the dimensions presented in the methodology: Supply Security; Scientific and technological support; Industrial resilience; International trade. To bring answers and comparisons with the data, however, other arguments emerged and the structure of the expected analysis changed.

In lines 508 to 532, the authors add an example of pork production, citing companies including, I do not understand the significance of the data at the company level, it is necessary to improve the explanation or withdraw the indication, because it does not run with the other productions.

In the conclusions,

Lines 680 and 700, both in second place? Do the authors indicate improvements and suggestions for China's livestock industry, based on the 4 dimensions and indicators in Table 1? Consider the alignment.

Under the theoretical aspect, what is the role of government and companies? What differs the established conditions in relation to other countries? What are the limitations of the analysis performed?

How does the study contribute to new research? What would be your suggestions for future research? I think it would be appropriate to point out some general aspects of world production and specific ones, as the analysis was developed.

I suggest the authors check the expression "construction of livestock powerhouses" the article compares livestock production data, but why establish powerhouses? I did not identify such usability of the expression in the literature.

Reviewer 2 Report

Dear authors! Your research is devoted to an interesting topic - the state of animal husbandry in China. A lot of work has been done, which leaves an ambiguous impression.

The article consists of two parts, loosely related to each other. The first, based on the documents of the Chinese Government, is the author's version of the ranking of world countries according to the state of animal husbandry. Unfortunately, the Chinese government is not a research organization. It is unjustified to expect scientific calculations from him and to build a scientific theory on them. Therefore, the result obtained in this part does not carry any scientific (reliable) information.

The second part of the article is a comparative analysis of the state of animal husbandry. This part of the study (unlike the first) clearly shows the limitations, shortcomings, and opportunities for animal husbandry in China. It is because of this part that I would like to give the authors the opportunity to print the results of their research.

Additionally, the article has more shortcomings.

The abstract in the article does not carry almost any semantic load. It does not reflect the content of the article.

Introduction. Very long, there are repetitions. References to literature sources fall on 1 paragraph of the introduction.

Methods and materials. Actually, methods account for 1 page out of 4. The remaining three pages are about nothing.

In Table 1, the choice of evaluation indices is ambiguous. The calculation of some indicators (for example, Revealed comparative advantage index) is not clear.

Based on this, the analysis made does not make sense, which is confirmed by further materials of the authors.

Discussion is good. The only remark is that this section usually does not present new information, but discusses the results obtained with references to the work of other authors.

Reviewer 3 Report

Dear Authors,

The article deals with an interesting topic for the Jounal, however there are some adjustments to be made, below you will find some general suggestions and other specific ones referring to single paragraphs.

Good work

In general:

·         Line 10: I think the sentence “These authors have contributed equally to this work” can be deleted.

·         The page numbering is wrong, please revise it.

·         The bibliography must be enriched because many statements are made, but without providing the bibliographic source.

·         Please explain acronyms.

·         International units of measure: it would be better to express the values ​​in dollars or in any case provide a dollar equivalent. As far as surfaces are concerned, it would be appropriate to use hectares.

·         In general, the text should be simplified to make it smoother for reading.

Introduction:

·         The objective of the work should be better specified, and the topic dealt with more clearly contextualized.

·         The document structure description is missing.

Materials and methods:

·         The methodological part should be simplified, instead of using so many subparagraphs, these parts could be reduced and become points in a bulleted list. The more extensive content of the current subparagraphs, on the other hand, could be moved to the introduction and in this way the reference framework would be clearer.

·         Equation 3 (line 245) shows the formula for k but does not explain what it represents.

Results and discussions:

·         Lines 288-291: I would suggest better structuring the comment, instead of introducing all the figures and tables first and then commenting on them, you could introduce each figure or table individually and comment on it directly.

·         Line 300-360: it would be advisable to specify which tables or figures the comments refer to, in order to facilitate reading.

·         Paragraph 3.2 the current discussions are actually a commentary on the results and therefore can remain in the text but without the need for an additional subparagraph. To have real discussions, a separate paragraph could be inserted in which the results obtained are compared with others present in the literature.

Paragraph 4:

·         It's not clear to me why, if you mention 135 countries then the comparison is made only with the USA, maybe I didn't understand but it should be better explained.

·         Table 6: the reference is missing in the text, please specify that it is China.

Conclusions: limits and future perspectives of the work should be highlighted

The quality of English Language could be improved. Please try to avoid repetitions and reformulate some sentences that at the moment seem to bee too long. 

Round 2

Reviewer 1 Report

After the adjustments made, I suggest that the authors review some aspects of the study presentation:

- long paragraphs in the study, can be improved, with adjustment of shorter sentences, division of paragraphs facilitate the reading and organization of the text.

- Observe phrases without the use of sources (missing authors) in some paragraphs of the introduction and theoretical review.

- Qualification of the analysis, there is still a lack of discussions with previous studies, to compare with data presented.

At this stage, I recommend reading and organizing the text.

Adequate.

Reviewer 2 Report

Dear authors! Thanks for good job.
